# Electrochemical Properties of an Sn-Doped LATP Ceramic Electrolyte and Its Derived Sandwich-Structured Composite Solid Electrolyte

**DOI:** 10.3390/nano12122082

**Published:** 2022-06-16

**Authors:** Aihong Xu, Ruoming Wang, Mengqin Yao, Jianxin Cao, Mengjun Li, Chunliang Yang, Fei Liu, Jun Ma

**Affiliations:** 1Department of Chemical Engineering, School of Chemistry and Chemical Engineering, Guizhou University, Guiyang 550025, China; 15851959055@163.com (A.X.); w1285459785@126.com (R.W.); mqyao@gzu.edu.cn (M.Y.); jxcao@gzu.edu.cn (J.C.); x1285459785@126.com (M.L.); clyang@gzu.edu.cn (C.Y.); 2Guizhou Key Laboratory for Green Chemical and Clean Energy Technology, Guiyang 550025, China

**Keywords:** NASICON-type LATP, Sn doping, sandwich structure, composite solid electrolyte, lattice distortion

## Abstract

An Li_1.3_Al_0.3_Sn_x_Ti_1.7−x_(PO_4_)_3_ (LATP-xSn) ceramic solid electrolyte was prepared by Sn doping via a solid phase method. The results showed that adding an Sn dopant with a larger ionic radius in a concentration of x = 0.35 enabled one to equivalently substitute Ti sites in the LATP crystal structure to the maximum extent. The uniform Sn doping could produce a stable LATP structure with small grain size and improved relative density. The lattice distortion induced by Sn doping also modified the transport channels of Li ions, which promoted the increase of ionic conductivity from 5.05 × 10^−5^ to 4.71 × 10^−4^ S/cm at room temperature. The SPE/LATP-0.35Sn/SPE composite solid electrolyte with a sandwich structure was prepared by coating, which had a high ionic conductivity of 5.9 × 10^−5^ S/cm at room temperature, a wide electrochemical window of 4.66 V vs. Li/Li^+^, and a good lithium-ion migration number of 0.38. The Li||Li symmetric battery test results revealed that the composite solid electrolyte could stably perform for 500 h at 60 °C under the current density of 0.2 mA/cm^2^, indicating its good interface stability with metallic lithium. Moreover, the analysis of the all-solid-state LiFePO_4_||SPE/LATP-0.35Sn/SPE||Li battery showed that the composite solid electrolyte had good cycling stability and rate performance. Under the conditions of 60 °C and 0.2 C, stable accumulation up to 200 cycles was achieved at a capacity retention ratio of 90.5% and a coulombic efficiency of about 100% after cycling test.

## 1. Introduction

At present, lithium-ion batteries have been widely used in various electronic products, electric/hybrid vehicles, and fixed energy storage systems [1,2,3,4]. However, the toxic organic liquid electrolytes commonly employed in traditional lithium-ion batteries have many shortcomings, such as undesirable inflammability, easy decomposition at high temperatures, rapid solidification at low temperatures, and fast leakage [5]. Moreover, side reactions of organic liquid electrolytes with positive and negative electrodes are prone to occur [6].

This issue can be successfully solved by introducing all-solid-state batteries in which organic liquid electrolytes can be replaced by inorganic solid electrolytes with high thermal stability [7,8]. In addition to safety, all-solid-state batteries also have many other advantages, such as simplified battery packaging, better electrochemical stability, and wider operating temperature ranges [6,9,10]. Inorganic lithium-ion solid electrolytes include NASICON-type [11,12], garnet-type [13], perovskite-type [14], LISICON [15], LiPON [16], Li_3_N [17], sulfides [18], and anti-perovskite [19] systems. Among them, NASICON-type solid electrolytes have attracted much attention due to their excellent electrochemical stability in air and/or water environment, as well as their low manufacturing cost [20,21].

Between various NASICON structures, LiTi_2_(PO_4_)_3_ (LTP) has a three-dimensional network assembled by two TiO_6_ octahedrons and three PO_4_ tetrahedrons sharing oxygen atoms, which can provide a three-dimensional interconnected conduction pathway for Li^+^ ion transport [22]. In particular, Al^3+^ doping of LTP (Li_1+x_Al_x_Ti_1.7−x_(PO_4_)_3_ or LATP) enables one to increase the carrier concentration and to reduce the Li-O bond strength, thus improving the ionic conductivity to a certain extent [23]. In recent years, ion doping has still been an effective strategy to upgrade the ionic conductivity of LATP. The LATP frameworks can be modified by intercalating cations with different valence states and ion radii, which causes lattice distortion and enhances ionic conductivity through the adjustment of ion transport channels and the increase of Li ion or Li vacancy concentration. For instance, doping modification of LATP by cations such as Nb^5+^ [24], Zr^4+^ [7], Y^3+^ [25], Ga^3+^ [26], Sc^3+^ [27], Te^4+^ [28], Si^4+^ [29], and V^5+^ [30] has been reported [31,32].

Nevertheless, the crystal structure of LATP after equivalent ion substitution for Ti^4+^ site (0.745 Å) exhibits the better structural stability. Doping modification with larger radius ions can also increase the lattice volume and widen the Li ion channels to a certain extent. In addition, introducing dopants with higher Pauli electronegativity allows one to improve the stability of cations in the LATP structure. Undoubtedly, the selection of dopants that are abundant, inexpensive, and eco-friendly is also a significant factor that cannot be ignored in doping modification.

Taking into account the above favorable factors, we performed doping modification of LATP by substituting Ti^4+^ (0.745 Å) with Sn^4+^ (0.83 Å) ions with larger ionic radius and higher Pauli electronegativity, in order to improve the framework stability and to make channel structures more conducive to ion diffusion. Because of its non-toxicity, low price, and abundant reserves [33], tin (Sn) has been widely concerned in photoelectric, photovoltaic, and energy storage devices [34,35,36,37], but its application in the modified LATP structures has not been reported yet.

Solid electrolytes are generally rigid and difficult to be machined [38]. When they contact with solid (either positive or negative) electrodes, there is often severe interfacial impedance [39]. Polymer solid electrolytes usually have good elastic properties and low interfacial impedance with solid electrodes [40,41]. The composite solid electrolyte formed by uniform mixing of polymer precursor solution and ceramic electrolyte powder can combine the respective advantages of both polymers and ceramics [42,43]. However, the reduction of the interfacial impedance between the composite solid electrolyte and the electrode is achieved at the expense of sacrificing conductivity of the ceramic electrolyte. This is because the conductivity of solid polymer electrolytes (SPE) represented by polyethylene oxide (PEO) is generally low (≤10^−6^ S/cm) [44,45,46].

Considering the above two aspects and focusing on the preparation of high-performance composite solid electrolytes, this paper places particular emphasis on the influence of Sn^4+^ doping on the structural properties and electrochemical performance of LATP. Attempts are made to reduce the solid-solid interface impedance through the construction of a sandwich-structured composite solid electrolyte by coating the PEO polymer electrolyte precursor solution on both sides of the modified ceramic electrolyte tablet instead of even mixing. The structure–property relationship of the composite solid electrolyte material was established by means of a series of analytical and electrochemical performance tests.

## 2. Materials and Methods

### 2.1. Materials

The raw materials included LiNO_3_ (99%, Aladdin, Shanghai, China), Al_2_O_3_ (99%, Aladdin, Shanghai, China), TiO_2_ (99%, Macklin, Shanghai, China), SnO_2_ (99.9%, Aladdin, Shanghai, China), NH_4_H_2_PO_4_ (99%, Macklin, Shanghai, China), Polyethylene Oxide (PEO) (Mw = 600,000, Macklin, Shanghai, China), and lithium bisimide (LiTFSI) (99.99% purity, Aladdin, Shanghai, China), which were not further purified.

### 2.2. Preparation of an LATP-xSn Ceramic Solid Electrolyte

The Sn-doped Li_1.3_Al_0.3_Sn_x_Ti_1.7−x_(PO_4_)_3_ (LATP-xSn, x = 0–0.4) solid electrolyte was prepared by a solid-phase reaction method. The stoichiometric quantities of raw materials for preparing an LATP-xSn ceramic solid electrolyte are as follows: LiNO_3_ (1.3), Al_2_O_3_ (0.3), SnO_2_ (x, where x = 0, 0.1, 0.2, 0.3, 0.35, 0.4), TiO_2_ (1.7 − x), and NH_4_H_2_PO_4_ (3) are mixed evenly in a ceramic mortar. Then, the powder mixture was calcined in an alumina crucible at 300 °C for 3 h. The temperature was then programmatically increased to 700 °C (5 °C/min) and held for 5 h. After cooling and grinding, the samples were pressed at 16 MPa for 10 min into wafers with a diameter of 13 mm and a thickness of 1.5–2.5 mm. Finally, the green-pressing wafers were sintered at 900 °C for 6 h.

### 2.3. Preparation of a Composite Solid Electrolyte

The PEO and lithium bisimide with a molar ratio of PEO:Li = 8:1 were dissolved in an acetonitrile solvent, and stirred for 12 h in a glovebox under an argon atmosphere to obtain a homogenous precursor solution of polymer electrolyte [47]. The solution was evenly coated on both sides of the LATP-xSn ceramic tablet, and then the sample was transferred to a vacuum oven for drying at 60 °C for 12 h to prepare an SPE/LATP-xSn/SPE sandwich-structured composite solid electrolyte.

### 2.4. Preparation of Electrodes and Assembly of the Battery

To fabricate a LiFePO_4_ cathode, the LiFePO_4_ powder, carbon black, and PVDF binder were mixed in N-methyl pyrrolidone at a mass ratio of 8:1:1 and stirred for 12 h to form a slurry of suitable viscosity, which was cast onto the surface of the current collector Al foil. The electrodes were placed in a blast drying oven at 60 °C for 12 h, and then transferred to a vacuum drying oven at 100 °C for 1 h. Afterwards, the dried pole piece was rolled until its surface was smooth and the thickness was 20 μm, and was punched into a circular pole piece with a diameter of 12 mm. The prepared polar pieces were dried in a vacuum drying oven for 12 h and then transferred to a glove box for use. The loading mass of the LiFePO_4_ in the electrode is 2.4 mg/cm^2^.

The above sandwich-structured composite solid electrolyte was assembled into a CR2032 button cell for an electrochemical performance test. LiFePO_4_ and lithium were used as the positive and negative electrodes, respectively, and the sandwich-structured composite solid electrolyte was placed between them (without any auxiliary electrolyte).

### 2.5. Materials Characterization

X-ray diffraction experiments were performed to identify the crystal structure of LATP-xSn (x = 0–0.4) on a Bruker D8 Advance X-ray diffractometer (XRD) equipped with a Cu Kα source (λ = 1.54178 Å) and operated at 40 kV and 40 mA. The XRD data were acquired in the 2θ range of 10–90° and were refined using FullProf software (FullProf Suit, (Version January-2021-JPC-JRC), signed with the “Institut Laue-Langevin” certificate, https://www.ill.eu/sites/fullprof/php/downloads.html (accessed on 11 August 2021)). The microstructure characterization and element distribution analysis were performed by means of a Carl Zeiss Supra 40 scanning electron microscope (SEM) and a Zeiss Gemini 300 energy dispersive X-ray spectrometer (EDS), respectively. The information about the molecular framework of LATP-xSn (x = 0–0.4) was extracted using a Thermo Fisher Nicolet IS50 Fourier transform infrared spectrometer and a LabRam HR Evolution laser microconfocal Raman spectrometer (Horiba Jobin Yvon, Paris, France). The surface elements and their valence states were analyzed on a K-Alpha Plus X-ray photoelectron spectrometer (Thermo Fisher, Waltham, MA, USA). The volumetric density of samples was measured using a Vernier caliper, and the theoretical density was obtained from the refined XRD data. The deviation degree (abbreviated as D) refers to the sum of the relative deviations of the ion radius and electronegativity between the dopant ion and the substituted ion. The calculation formula is as follows [29]:(1)$$D=\left|\frac{\chi_{Sn}-\chi_{i}}{\chi_{i}}\right|+\left|\frac{\gamma_{Sn}-\gamma_{i}}{\gamma_{i}}\right|$$
where $\chi_{Sn}$ and $\gamma_{Sn}$ are the ion electronegativity and crystal ion radius of Sn^4+^, respectively, and $\chi_{i}$ and $\gamma_{i}$ are the ion electronegativity and crystal ion radius of the substituted ion, respectively.

### 2.6. Electrochemical Performance Test

The ionic conductivity, electrochemical stability window, and lithium-ion migration number of the solid electrolyte were measured using a CHI660E (Chenhua, Shanghai, China) electrochemical workstation.

Ionic conductivities were measured by electrochemical impedance spectroscopy with a signal amplitude of 5 mV over the frequency range of 10^−2^–10^6^ Hz at 25 °C. The ionic conductivity σ (S/cm) is calculated as follows:(2)σ=L/RS
where *L* (cm) is the thickness of the electrolyte, *R* (Ω) is the resistance value of the electrolyte, and *S* (cm^2^) is the area of the electrode plate.

By testing the ionic conductivities at different temperatures, the conductive activation energy *Ea* (eV) of Li^+^ can be calculated from the Arrhenius formula as follows:(3)$$\sigma T=Aexp\left(-\frac{Ea}{K_{B}T}\right)$$
where *σ* is the ionic conductivity of the electrolyte, *T* is the temperature, *A* is the pre-exponential factor, and *K_B_* is the Boltzmann’s constant.

Linear sweep voltammetry (LSV) was performed to investigate the electrochemical stability window of the solid electrolyte with a scanning rate of 0.001 V/s at 60 °C from 2 V to 6 V (vs. Li^+^/Li). The lithium ion transport numbers ($t_{Li^{+}}$) of the solid electrolyte were measured with a DC polarization voltage of 10 mV associated with the AC impedance measurement and calculated using Equation (4):(4)$$t_{Li^{+}}=I_{SS}\left(\Delta V-I_{0}R_{0}\right)/(I_{0}\left(\Delta V-I_{ss}R_{ss}\right)$$
where $I_{0}$ and $I_{SS}$ are the initial current and steady-state current, both furnished by a direct current (DC) polarization test, $R_{0}$ and $R_{SS}$ are the charge transfer resistances before and after DC polarization, and ΔV is the polarization voltage.

The Li||Li symmetric constant current charge and discharge curves of the composite solid electrolyte, as well as the cycle performance and rate performance of the all-solid-state battery, were measured using a CT3001A battery testing system (Lanhe, Wuhan, China).

## 3. Results

### 3.1. Influences of Sn Doping on the Structural Properties and Electrochemical Performance of an LATP-xSn Solid Electrolyte

Figure 1 shows the XRD patterns of a series of modified LATP-xSn solid electrolytes (x = 0–0.4). The diffraction peaks of the main crystal phase of all modified samples are basically consistent with those of a LiTi_2_(PO_4_)_3_ (PDF#35-0754) with an R-3c space group, indicating the successful preparation of a Li_1.3_Al_0.3_Sn_x_Ti_1.7−x_(PO_4_)_3_ solid electrolyte with a NASICON-type structure. However, a small amount of AlPO_4_ crystalline phase appears in the LATP-0.1Sn and LATP-0.2Sn samples. With the increase of Sn content, the AlPO_4_ phase gradually disappears. Moreover, the LATP-0.4Sn sample reveals the emergence of an SnO_2_ phase. It should be pointed out that the diffraction peak associated with a (113) crystal plane of the modified samples tends to shift to the lower-angle range, and the shift degree becomes more pronounced with the increase of Sn dopant content. This trend can be explained by the effective substitution of smaller ions in the LATP lattice by the larger Sn^4+^ (r = 0.083 nm) ions.

This speculation can be further confirmed by the lattice parameters of the refined data (Table 1, Appendix A). The lattice parameters of modified LATP-xSn (x = 0–0.35) systems increase linearly with the increase of Sn content, which follows the Vegard law, indicating an effective insertion of Sn^4+^ ions [48]. It has been reported that the increase of lattice parameters can facilitate the intercalation and deintercalation of lithium ions during the cycling process [49]. However, the lattice parameters in a and b axes of the modified LATP-0.4Sn exhibit a decreasing trend, which may be related to the appearance of the SnO_2_ phase in the sample [48]. The theoretical density value of the modified samples also shows a similar variation trend. Rp is the abbreviation of R-pattern, which means graphic variance factor. Rwp is the abbreviation of R-pattern, which means weighted graphic variance factor. Rexp is the abbreviation of R-expected, which means expected variance factor. These are fitting factors, used to judge the result of refinement.

It is generally believed that doping is more likely to occur at ion sites of similar electronegativity and ion radiuses [50,51]. Thus, the spatial effect and electrostatic interaction can be evaluated from the ion radius and ion electronegativity, respectively [52], and the calculated deviation degree can reflect the most probable Sn doping sites in the lattice. The deviation degree results (Appendix A) calculated according to Equation (1) show that the deviation degree value of DTi is the smallest, indicating that Sn is more inclined to occupy octahedral Ti sites in the LATP lattice. This speculation can be verified by the following characterization results.

The Raman-active vibration modes of the LATP-xSn (x = 0–0.4) solid electrolyte are usually divided into internal and external modes. Internal modes occur above ~350 cm^−1^ in the Raman spectra. In Figure 2, the peak at 350 cm^−1^ is ascribed to the Ti-O vibration mode [50], the signal peaks at 435 and 450 cm^−1^ can be attributed to the symmetric bending motion of P-O bonds in PO_4_ tetrahedrons [51], and the signal peaks at 971, 990, and 1009 cm^−1^ correspond to asymmetric and symmetric stretching vibrations of the P-O bonds [50]. Except for the LATP-0.4Sn sample, all peak intensities of other samples increase with the increase of Sn doping content. Once the Sn content increases to x = 0.35, the peak intensities increase to their maximum values. It is worth mentioning that the band width of the Ti-O peak at 350 cm^−1^ increases after Sn doping, indicating that the lattice distortion around TiO_6_ is enhanced due to the substitution of Ti sites by Sn ions [53].

In addition, the relative intensities of the peaks at 350 and 314 cm^−1^ in Figure 2 can reflect the alteration of the lithium content in the LATP lattice. When the lithium content is high, the peak intensity at 314 cm^−1^ is higher than that at 350 cm^−1^, and vice versa [54]. Except for the LATP-0.4Sn sample, the peak intensity at 314 cm^−1^ gradually increases with the increase of Sn dopant content, indicating that Sn doping can reduce lithium loss. The largest intensity difference between the two peaks is obtained at the lithium content of x = 0.35.

Figure 3 displays the infrared spectra of the LATP-xSn (x = 0–0.4) solid electrolyte. In the figure, the peaks at 580 and 648 cm^−1^ can be attributed to the Ti-O stretching vibrations of TiO_6_ octahedra [55], which overlap with the Sn-O vibrations of SnO_6_ octahedra at 648 cm^−1^ [56]. With the increase of Sn dopant content, the peak intensity at 648 cm^−1^ obviously increases because Sn doping promotes the enhancement of the Ti-O bond strength. This change in structural information can also be verified by XPS results (Appendix A). The characteristic peaks at 487.3 and 495.7 eV in the LATP-0.35Sn sample confirm the successful doping by Sn^4+^ ions. In addition, the shifts of Ti 2p and O 2p characteristic peaks confirm that the O-Sn bonds in Sn-doped LATP are formed by the substitution of Sn for Ti.

The electrochemical impedance spectra of the LATP-xSn (x = 0–0.4) solid electrolyte were measured (Figure 4a). The semicircle displayed in the high frequency region is related to the conductivity of the solid electrolyte, and the straight line in the low frequency region represents the resistance of lithium ion transport between the electrolyte and the blocking electrode. The high frequency region does not display the grain resistance (Rg) due to the limitation of the test frequency. The grain resistance (Rg) and grain boundary resistance (Rgb) can be obtained by fitting the equivalent circuit with Zview software, and CPE is a constant phase element, which is used to replace capacitor in many equivalent circuit models. Rg, Rgb, CPE1, and CPE2 represent grain resistance, grain boundary resistance, grain boundary capacitance, and sample-electrode capacitance respectively. The total resistance (Rt) is the grain resistance (Rg) and the grain boundary resistance (Rgb) [57]. The relevant ionic conductivity data were calculated according to Equation (2) (Figure 4a and Table 2). Due to the low sintering temperature of LATP solid electrolyte, the total ionic conductivity of LATP before being undoped is low (5.05 × 10^−5^ S/cm), which is significantly lower than the reported value in the previous work [48,55]. The room temperature ionic conductivity of LATP-0.35Sn is 4.71 × 10^−4^ S/cm, which is nearly an order of magnitude higher than that of the undoped one. According to Table 2, it is found that the total ionic conductivity of LATP-0Sn is mainly determined by the ionic conductivity of the grain boundary. With the increase of the Sn content, the grain boundary impedance is greatly reduced, and the total ionic conductivity is significantly improved. The ionic conductivity is the highest (4.71 × 10^−4^ S/cm), which is higher than that of other typical solid electrolytes (Appendix A) [8,58,59,60,61,62]. However, when the Sn content reaches 0.4, both the grain conductivity and grain boundary conductivity decrease.

The electrochemical impedance spectra of the LATP-xSn (x = 0–0.4) solid electrolyte were tested in the temperature range of 25–125 °C, and the conductive activation energy of lithium ions was calculated according to Equation (3) (Figure 4b and Table 2). All samples show good linear fitting results, among which LATP-0.35Sn exhibits a minimum activation energy of 0.23 eV and a relatively high relative density (91.8%), indicating that Li^+^ ions can migrate more easily in the LATP-0.35Sn lattice and through the grain boundaries. When the Sn doping content is x = 0.4, the relative density continues to increase to 92.0%, but the ionic conductivity decreases, which is related to the presence of the SnO_2_ phase in the sample [52].

Through the equivalent substitution of Ti^4+^ sites by Sn^4+^, Sn ions can occupy the LATP lattice uniformly (Appendix A), and the crystal structure does not experience neither excessive lattice distortion nor crystal transformation. More importantly, smaller grains can be obtained (Appendix A) and the relative density is also improved (Table 2), finally enhancing the ionic conductivity.

### 3.2. Electrochemical Performance of the Composite Solid Electrolyte

The solid–solid interface between the LATP-0.35Sn solid electrolyte and lithium has a serious interface impedance problem, and the ionic conductivity of the LATP-0.35Sn all-solid-state battery at 25 °C is only 1.0 × 10^−5^ S/cm (Appendix A). The thickness of the LATP-0.35Sn solid electrolyte is 1.2 mm. After coating the surface with SPE, the SPE film is smooth and dense (Appendix A), and the thickness of the film is about 24 μm on one side (Appendix A). The electrolyte resistance (Re) and charge transfer resistance (Rct) were obtained by fitting the equivalent circuit through the software Zview (Appendix A). The interface impedance of the SPE/LATP-0.35Sn/SPE composite solid electrolyte and the electrode was significantly reduced, and its charge transfer resistance Rct (1344 Ω) is much lower than the LATP-0.35Sn charge transfer resistance Rct (8541 Ω), and the difference between the two lies in the PEO-LiTFSI on the surface of SPE/LATP-0.35Sn/SPE. The charge transfer resistance Rct (4408 Ω) of PEO-LiTFSI is much lower than that of the LATP-0.35Sn charge transfer resistance Rct (8541 Ω), indicating that it has better contact with electrodes and is more favorable for charge transfer between electrodes. The electrolyte resistance Re (239 Ω) of SPE/LATP-0.35Sn/SPE is lower than that of LATP-0.35Sn (413 Ω) and that of PEO-LiTFSI (1739 Ω), which is attributed to the fact that PEO-LiTFSI is filled with pores on the surface of the LATP-0.35Sn solid electrolyte and provides more Li^+^ diffusion pathways for solid–solid interfaces. A salt anion TFSI-LATP-0.35Sn surface with stronger affinity accelerates lithium ion migration [63]. Therefore, the room temperature ionic conductivity of SPE/LATP-0.35Sn/SPE was improved to 5.9 × 10^−5^ S/cm (Appendix A).

The impedance spectra before and after polarization and DC polarization curves of the SPE/LATP-0Sn/SPE and SPE/LATP-0.35Sn/SPE systems at 60 °C were measured for comparison (Figure 5), and the migration number of Li^+^ ($t_{Li^{+}}$) were calculated according to Equation (4) (Table 3). As can be seen, the migration number of Li^+^ ions for SPE/LATP-0Sn/SPE is only 0.27, whereas that of the SPE/LATP-0.35Sn/SPE is increased to 0.38. Combining the LATP-0.35Sn solid electrolyte with high ionic conductivity and the polymer can help to fix TFSI^−^ and release more Li^+^ ions for transport, thus increasing the migration number of Li^+^ [64]. As a result, the increase of $t_{Li^{+}}$ enables one to reduce the polarization of the all-solid-state battery and promote the uniform deposition of Li [63].

The LSV curves of PEO-LiTFSI, SPE/LATP-0Sn/SPE, and SPE/LATP-0.35Sn/SPE at 60 °C are given in Figure 6. The electrochemical window of PEO-LiTFSI is only 3.76 V vs. Li/Li^+^. This is mainly attributed to the structural instability of the pure PEO-LiTFSI polymer electrolyte in the electric field, which is prone to the electrochemical oxidation decomposition [65]. The electrochemical window of SPE/LATP-0Sn/SPE is 4.34 V, whereas that of SPE/LATP-0.35Sn/SPE increases to 4.66 V. These mean that the designed sandwich composite structure can not only effectively hindering the transfer of TFSI^−^ anions and thus reduce the decomposition of PEO, but also block the direct contact between the metallic Li negative electrode and the solid electrolyte, avoiding the occurrence of side reactions [66]. Moreover, the proposed sandwich structure makes full use of the advantage of the high electrochemical performance of an LATP-0.35Sn.

Figure 7 displays the electrochemical compatibility and stability results of SPE/LATP-0Sn/SPE and SPE/LATP-0.35Sn/SPE composite solid electrolytes and metallic lithium. The voltage of the SPE/LATP-0Sn/SPE begins to increase gradually after 26 h of cycling at a current density of 0.5 mA/cm^2^, and exceeds the safety voltage of 5 V after 100 h. By contrast, the SPE/LATP-0.35Sn/SPE can perform stably for 500 h at a current density of 0.2 mA/cm^2^ and show a low polarization voltage (~59 mV), indicating that the migration resistance of lithium ions between the composite solid electrolyte and Li metal is low. The polarization voltage of the lithium symmetric battery decreases slightly during the first cycles at the current density of 0.5 mA/cm^2^. After cycling for 23 h, the polarization voltage of the lithium symmetric battery enhances stability, which may be due to the interface optimization caused by repeated electroplating/stripping of lithium [67]. At a higher current density of 0.5 mA/cm^2^, the battery can still circulate for 300 h, and the voltage slightly changes with time. On the whole, the SPE/LATP-0.35Sn/SPE electrolyte exhibits good stability of the metallic Li interface, which seems to be due to the stable contact at the PEO/Li interface. More importantly, the compact structure of the modified LATP−0.35Sn solid electrolyte can enhance the mechanical strength and increase the ionic conductivity of PEO. In addition, the modified LATP-0.35Sn solid electrolyte can make the distribution of lithium ions more uniform and thus effectively prevent the growth of lithium dendrites.

The sandwich-structured SPE/LATP-0.35Sn/SPE composite solid electrolyte, the cathode of the lithium iron phosphate (LiFePO_4_), and the anode of the metallic lithium (Li) were afterwards assembled into an all-solid-state battery. The cycling performance and rate performance of the LiFePO_4_||SPE/LATP-0.35Sn/SPE||Li battery were tested, and the results are shown in Figure 8. At a current density of 0.2 C, the all-solid-state battery can perform stably up to 200 cycles; the battery capacity decreases from 153.5 to 138.9 mAh/g, with a capacity retention rate of 90.5%, and the coulombic efficiency is close to 100% (Figure 8a,b). Furthermore, the LiFePO_4_||SPE/LATP-0.35Sn/SPE||Li all-solid-state lithium battery can also be cycled stably for 100 cycles at the high rate of 0.5 C. The battery capacity is reduced from 145.1 mAh/g to 132.3 mAh/g with a capacity retention rate of 91.2%, and the coulombic efficiency is close to 100% (Figure 8c). As the number of charge-discharge cycles increases, the voltage and the interface impedance change slightly (Figure 8d,e), indicating that the electrode/electrolyte interface is stable in the long-term cycling process. Figure 8f exhibits the rate performance of the all-solid-state battery in the range from 0.1 to 2 C for every five cycles. The first-cycle discharge capacity of the all-solid-state battery is 155.4 mAh/g at a low rate current of 0.1 C, and decreases to 112.7 mAh/g at a higher rate current of 2 C. After cycling at different rate currents, when the rate current returns to 0.2 C, the discharge capacity recovers a value of 152.4 mAh/g. The excellent rate performance is attributed to the good compact structure of the solid-phase interface layer and its ability of uniformly depositing lithium ions. Therefore, the SPE/LATP-0.35Sn/SPE composite solid electrolyte has excellent cycling performance and high rate charge-discharge characteristics.

## 4. Conclusions

A Sn-doped NASICON-type LATP ceramic solid electrolyte was prepared by the solid-phase method. The influences of different Sn dopant contents on the structural properties and electrochemical performance of the LATP solid electrolyte were investigated. After Sn doping, the ionic conductivity of Li_1.3_Al_0.3_Sn_0.35_Ti_1.35_(PO_4_)_3_ (LATP-0.35Sn) at room temperature could reach 4.71 × 10^−4^ S/cm, which was mainly attributed to the fact that Sn^4+^ with a larger ionic radius could substitute the Ti sites in the LATP crystal structure to the maximum extent. The uniform Sn distribution was conducive to the structural stability and the decrease in grain size of the crystal, thereby improving the relative density. Moreover, the lattice distortion caused by Sn doping also modified the transport channels of Li ions. The prepared sandwich-structured SPE/LATP-0.35Sn/SPE composite solid electrolyte exhibited good electrochemical performance, with an ionic conductivity of 5.9 × 10^−5^ S/cm at room temperature, an electrochemical stability window of 4.66 V vs. Li/Li^+^, and a lithium-ion migration number of 0.38. The SPE/LATP-0.35Sn/SPE composite solid electrolyte was afterwards used to assembly a Li||Li symmetric battery, revealing stable cycling performance for 500 h at 60 °C under the current density of 0.2 mA/cm^2^. Additionally, combining the SPE/LATP-0.35Sn/SPE composite solid electrolyte with a LiFePO_4_ cathode and a metallic Li anode enabled one to obtain an all-solid-state battery with excellent cycling stability and rate performance. The capacity retention rate of the all-solid-state battery cycled at a low rate of 0.2 C and at high rate of 0.5 C, reaching 90.5% and 91.2%, respectively. The coulombic efficiency at different rates are close to 100%.

## Figures and Tables

**Figure 1 nanomaterials-12-02082-f001:**
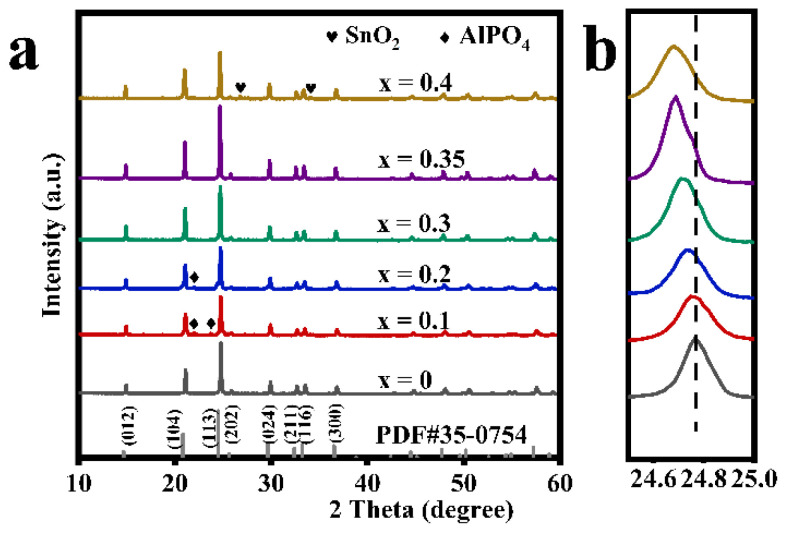
(**a**) X-ray diffraction patterns of LATP-xSn (x = 0–0.4) solid electrolytes and (**b**) the magnified view of diffraction patterns at (113) plane for LATP-xSn (x = 0–0.4) solid electrolytes.

**Figure 2 nanomaterials-12-02082-f002:**
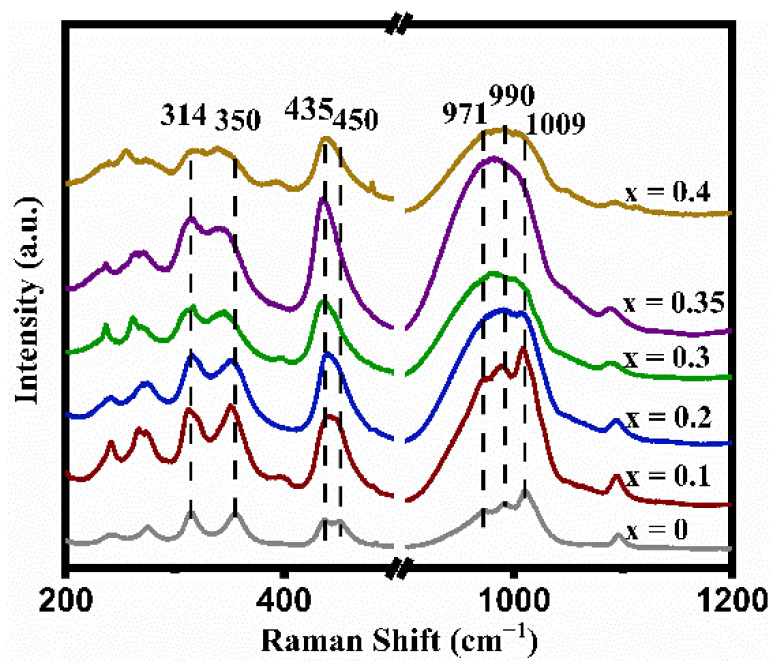
Raman spectra of LATP-xSn (x = 0–0.4) solid electrolytes.

**Figure 3 nanomaterials-12-02082-f003:**
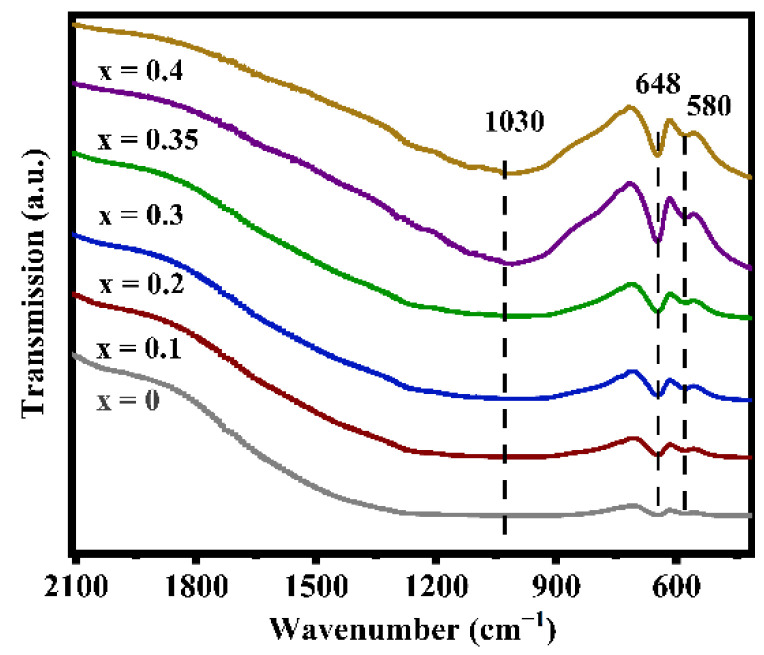
FT-IR spectra of LATP-xSn (x = 0–0.4) solid electrolytes.

**Figure 4 nanomaterials-12-02082-f004:**
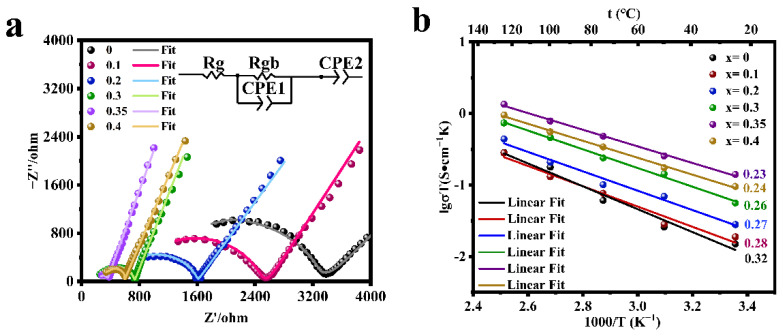
(**a**) EIS curves of LATP-xSn (x = 0–0.4) solid electrolytes at 25 °C and (**b**) the Arrhenius plot of the lithium ionic conductivity of LATP-xSn (x = 0–0.4) solid electrolytes.

**Figure 5 nanomaterials-12-02082-f005:**
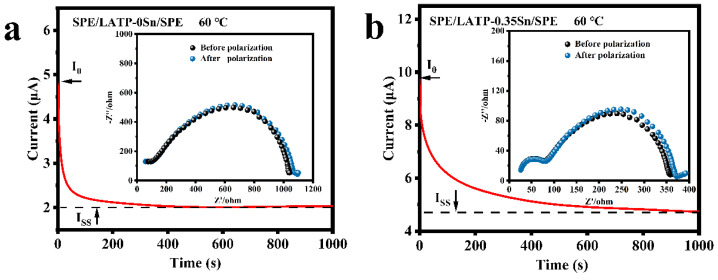
Impedance spectra and DC polarization for (**a**) SPE/LATP-0Sn/SPE and (**b**) SPE/LATP-0.35Sn/SPE.

**Figure 6 nanomaterials-12-02082-f006:**
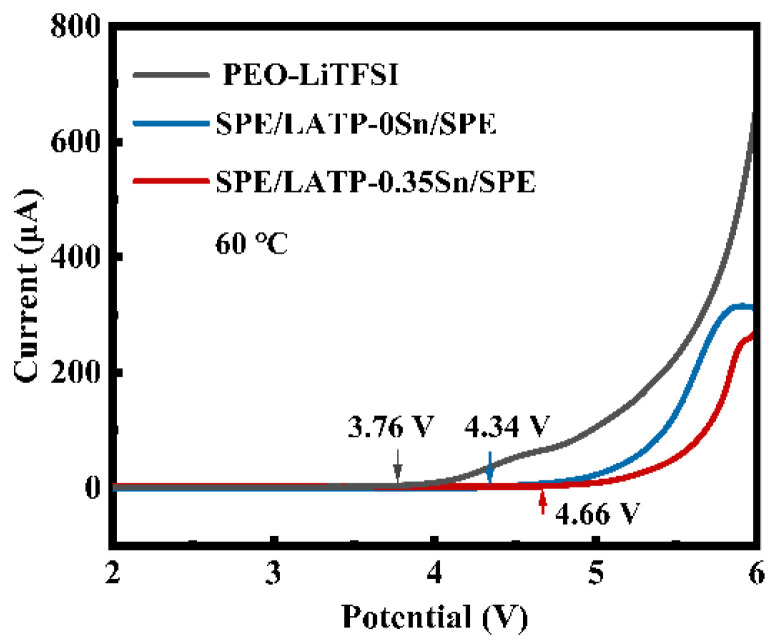
LSV curves of PEO-LiTFSI, SPE/LATP-0Sn/SPE, and SPE/LATP-0.35Sn/SPE solid electrolytes.

**Figure 7 nanomaterials-12-02082-f007:**
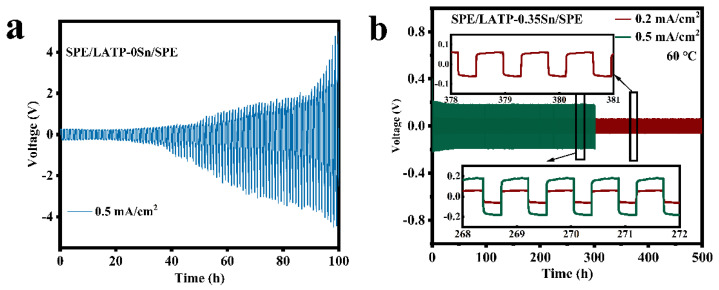
Voltage of the Li||Li symmetrical cells with (**a**) SPE/LATP-0Sn/SPE at the current density of 0.5 mA/cm^2^ and (**b**) SPE/LATP-0.35Sn/SPE at the current density of 0.2 mA/cm^2^ and 0.5 mA/cm^2^.

**Figure 8 nanomaterials-12-02082-f008:**
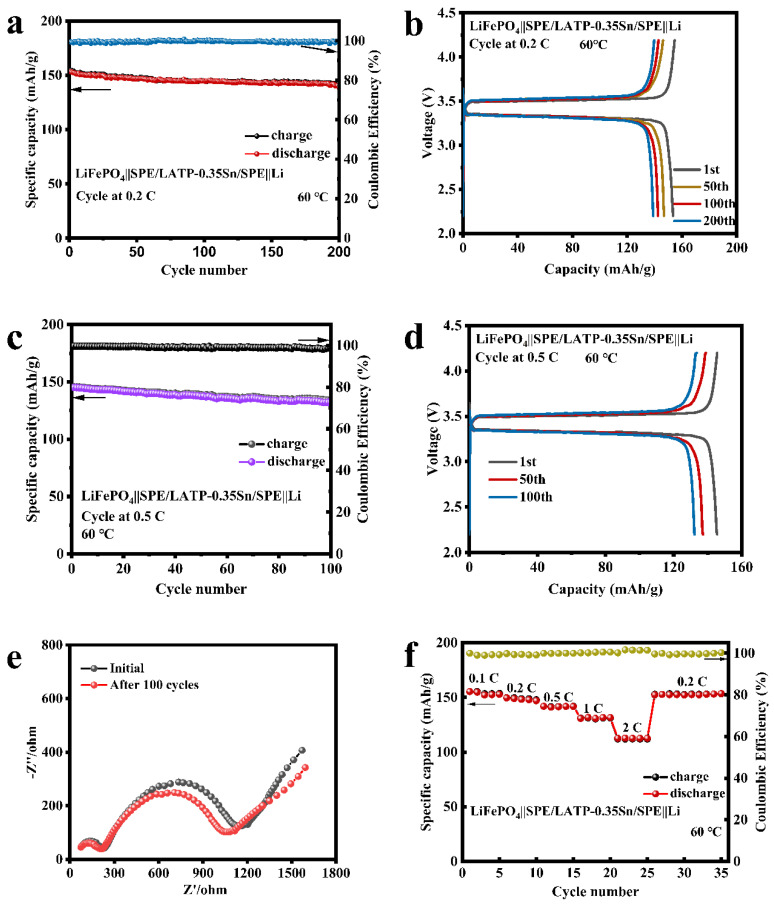
Electrochemical performance of LiFePO4||SPE/LATP-0.35Sn/SPE||Li all-solid-state lithium battery: (**a**) cycle performance at 0.2 C; (**b**) charge-discharge curves at 0.2 C; (**c**) cycle performance at 0.5 C; (**d**) charge-discharge curves at 0.5 C; (**e**) Electrochemical impedance plots before and after cycling at a rate of 0.5 C; (**f**) rate performance at various rates ranging from 0.1 C to 2 C.

**Table 1 nanomaterials-12-02082-t001:** Lattice parameters of LATP-xSn (x = 0–0.4) solid electrolytes.

x	a = b (Å)	c (Å)	Density (g/cm^3^)	Rp	Rwp	Rexp
0	8.509	20.864	2.92	9.8	13.5	10.3
0.1	8.515	20.884	2.78	10.1	14.2	9.2
0.2	8.522	20.919	2.95	11.2	13.9	10.9
0.3	8.526	20.939	3.31	9.3	13.1	10.2
0.35	8.528	20.944	3.32	10.3	14.0	11.5
0.4	8.525	20.973	3.11	9.4	13.3	10.5

**Table 2 nanomaterials-12-02082-t002:** Grain impedance (Rg), grain boundary impedance (Rgb), grain conductivities (σg), grain boundary conductivities (σgb), total conductivities (σt), activation energy (Ea), and relative density of LATP-xSn (x = 0–0.4) solid electrolytes.

x	Rg (Ω)	Rgb (Ω)	σg (mS/cm)	σgb (mS/cm)	σt (mS/cm)	Ea (eV)	Relative Density (%)
0	653	2798	0.267	0.0624	0.0505	0.32	87.5
0.1	580	1956	0.277	0.0821	0.0633	0.28	88.2
0.2	375.2	1237.4	0.402	0.122	0.0939	0.27	89.4
0.3	232	498.8	0.595	0.277	0.189	0.26	90.1
0.35	218.5	139.6	0.772	1.21	0.471	0.23	91.8
0.4	339.5	253.9	0.556	0.742	0.322	0.24	92.0

**Table 3 nanomaterials-12-02082-t003:** Lithium ion transport numbers ($t_{Li^{+}}$) of composite solid electrolytes at 60 °C.

Electrolytes	$I_{0}\;(\mu A)$	$I_{ss}\;(\mu A)$	$R_{0}\;(\Omega )$	$R_{ss}\;(\Omega )$	$t_{Li^{+}}$
SPE/LATP-0Sn/SPE	4.79	2.03	1039.8	1084.1	0.27
SPE/LATP-0.35Sn/SPE	9.72	4.73	358.2	372.3	0.38

## Data Availability

The data that support the findings of this study are available upon reasonable request.

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
