# Peer review of "Electrochemical Properties of an Sn-Doped LATP Ceramic Electrolyte and Its Derived Sandwich-Structured Composite Solid Electrolyte"

_nanomaterials, 2022, doi:10.3390/nano12122082_

Round 1

Reviewer 1 Report

The manuscript reports the preparation of Sn-doped LATP ceramic electrolyte and its derived sandwich-structured composite solid electrolyte. The results are interesting; however, still some issues need to be clarified before it can be published.

  1. In the preparation of LATP-xSn ceramic solid electrolyte, the stoichiometric quantities of the materials mixed should be provided.
  2. Preparation of composite solid electrolyte: The PEO (Mw = 600000, Macklin) and lithium bisimide (LiTFSI), 99.99% purity, Aladdin) with a molar ratio of EO:Li = 8:1 were dissolved in an acetonitrile solvent. Is it EO? Please correct it.
  3. From the Figure 8f, Can author explain why the specific capacity at 0.2C after 2C is more than its initial value(at 0.2C)?
  4. Define Rp, Rwp, and Rexp in the manuscript.
  5. The authors should give a table to compare the property/performance of the as-obtained LATP-xSn with other typical solid electrolytes that have been reported in the literature.
  6. The following related literatures on solid-state electrolytes and LFP cathode should be cited properly, e.g., (Nanomaterials 12, no. 7 (2022): 1158, Nano Energy, 2017, 33, 363-386. Materials Chemistry Frontiers, 2021, 5, 5336-5343. Electrochimica Acta 310 (2019): 222-229, Materials Today Communications 23 (2020): 100926, Journal of Alloys and Compounds 872 (2021): 159719)

Reviewer 2 Report

Manuscript tittled “Electrochemical properties of a Sn-doped LATP ceramic electrolyte and its derived sandwich-structured composite solid electrolyte” by A. Xu et al shows the synthesis and characterization of Li1.3Al0.3SnxTi1.7-x(PO4)3 solid solution (x=0-0.5, LATP-xSn) obtained by solid state procedure. Characterization included structural, spectroscopic, electrical and electrochemical testing to stablish the performance and potential application of such series as electrolyte for solid-state-lithium batteries.

The manuscript is based on the very well-known strategy to improve the electrochemical performance of LATP by iso- or aliovalent doping on Ti4+ site. There are significant research related to this item but for now, the substitution of Sn4+ ion by Ti4+ does not been already explored according to bibliography. Therefore, this is an interesting research in spite of the closely related previous studies.

Otherwise, next comments should be properly solved or commented by the authors:

-         - page 3, Experimental section; ”… with a molar ratio of EO:Li = 8:1…”, maybe is just a mistake and EO refers to PEO.  

-          -Page 4, Experimental section; regarding the electrical and electrochemical experimental conditions, it should be added temperature range, frequency range and applied voltage in EIC experiment as well as polarization voltage, DV, in DC polarization test.

-          -Page 5; “…follows the Veigard law…” must be corrected to “Vegard Law” (or Vegard’s Law).

-          -Page 5; “…The values of Rp, Rwp, and Rexp are all less than 15, meaning the reliability of the refinement results”. This is a quite hazardous affirmation and probably is not true. There is quite discussion regarding this point (how good would be R factor in Rietveld analysis to be good enough?). Currently, generally speaking, the most significant trend to assess the quality of a Rietveld fitting is the graphical checking of the difference between the observed and calculated patterns in order to ensure that the proposed structural model is chemically feasible. If we consider just the chi-square values and the R factor obtained in this research, higher than 10%, this would suggest us about a problem with the proposed model; consider, for example that it is not possible to make a clear distinction between R-3c and R3c symmetry groups using powder diffraction (more precise techniques such TEM and single-crystal diffraction must be employed. Authors should look up Powder Diffraction, 21 (1), 2006, pp. 67 – 70 for a highly authorized information about this.

-          -Page 5: following with the structural characterization, some additional diffraction peaks are found in Fig. S1d and f, corresponding to LATP-xSn (x= 0.3 and 0.4) at almost 2theta = 20° and 25°. Maybe the first one is related with SnO2 additional phase, but it is not clear at all that the second one (clearly highlighted in the difference pattern) is due to this secondary and phase. Moreover, if this is related to the secondary phase, the isovalent substitution higher limit may be compromised at x=0.35, being lower, thus.

-          Page 6-7; “With the increase of Sn content…the grain boundary resistance decreases obviously”. Check the meaning of this sentence, it is difficult to understand.  

-          Pages  6 and 7; according to the EIC proposed model, R1 must be identified with Rg and R2 with Rgb. This is not clearly stated in the manuscript as well as the meaning of CPE1 (this confusing because of the same index than for R1) nor the capacity values to support such matching. Therefore, model must be explained and physically identified more properly than it is.

-          Page 8, line 289; what is ”para-lithium”?

-          Page 8, line 294; “…(Figure 4)…” must be “… (Figure 5)…”

-          Page 9, Figure 7b; it is possible to observe an improvement of polarization voltage at 0.5 mA/cm2. This should be discussed, maybe in terms of compatibility and interface improvement during the first cycles.

According to these comments, I consider this manuscript suitable for publication in Nanomaterials, as although must be fulfilled by the authors, most of them are just minor corrections.

Reviewer 3 Report

The manuscript reports the structural and electrochemical characterisation of LATP solid electrolyte containing Sn additions. This characterisation appears relevant to the field, but this manuscript entirely does not fall into the scope of Nanomaterials journal, namely, to publish “reviews, regular research papers, communications, and short notes that are relevant to any field of study that involves nanomaterials, with respect to their science and application”. This study does not involve nanomaterials or any characterisation at the nanoscale. Thus, the manuscript is recommended for rejection and resubmission to a more suitable journal (e.g., Materials). The manuscript would further benefit from a more careful analysis of the XRD patterns, which suggest the presence of impurities already for x=0.1, as evidenced by small peaks between 2-25º. More detailed microstructural analysis by SEM/EDS is also recommended. The images shown in the Supplementary Materials section are taken at relatively high resolution and do not allow to see a more general picture, evaluate the quality of the electrolyte and compare the samples between them.

Round 2

Reviewer 3 Report

The authors did not address the comments carefully. All additional peaks must be indexed on the XRD pattern; corresponding clarifications must be added to the manuscript text.

The authors state: “The characterization by SEM and EDS (Figure R1 and R2) clearly shows microstructures of the solid electrolytes, which observe that the solid electrolytes are composed of nano particles. Therefore, the manuscript falls into the scope of Nanomaterials journal.” The images R1 and R2 do not show clearly any nanoparticles, rather highlighting the grains with a size certainly larger than 1 micrometer. What is probably considered “nanoparticles” there is just smaller particles or even dust present at the grain boundaries and as inclusions. Such submicrometric inclusions can be found almost in every ceramic material prepared by the conventional solid-state route without involving any method which is specific to nanomaterials. Thus, I still believe that this manuscript does not fall into the scope of Nanomaterials journal.

No additional SEM/EDS studies allowing to see a more general picture, evaluate the quality of the electrolyte and compare the samples between them were performed.

Based on the comments above, the manuscript is recommended for rejection.

Author Response

We thank the reviewer for pointing out this good question. The Figures R1 and R2 show dense solid materials formed by pressing nanomaterials under high pressure or after a certain heat treatment process, which belong to the nanostructured solid bulk materials. Also nanostructured solid state bulk materials fall into the scope of Nanomaterials. This manuscript is about the application of nanostructured morphological materials prepared as solid electrolytes in all-solid-state lithium-ion batteries and then falls within the scope of the Nanomaterials journal.
